# Assessment of Knowledge, Attitudes, and Practices Regarding Dengue among Physicians: A Web-Based Cross-Sectional Survey

**DOI:** 10.3390/bs11080105

**Published:** 2021-07-21

**Authors:** Kranthi Swaroop Koonisetty, Nasrin Aghamohammadi, Tamanna Urmi, Sare İlknur Yavaşoglu, Md. Shahinur Rahman, Rajesh Nandy, Ubydul Haque

**Affiliations:** 1Department of Biostatistics and Epidemiology, University of North Texas Health Science Center, Fort Worth, TX 76107, USA; Rajesh.Nandy@unthsc.edu (R.N.); Mdubydul.Haque@unthsc.edu (U.H.); 2Centre for Epidemiology and Evidence-Based Practice, Department of Social and Preventive Medicine, University of Malaya, Kuala Lumpur 50603, Malaysia; nasrin@ummc.edu.my; 3Department of Epidemiology, National Institute of Preventive and Social Medicine (NIPSOM), Dhaka 1212, Bangladesh; tamanna145urmi@gmail.com; 4Department of Biology, Faculty of Science and Arts, Aydın Adnan Menderes University, Aydın 09010, Turkey; sarecihangir@adu.edu.tr; 5Palli Karma Sahayak Foundation (PKSF), PKSF Bahban, E-4/B, Agargaon A/A, Sher-e-Bangla Nagar, Dhaka 1207, Bangladesh; kabires9@yahoo.com

**Keywords:** *Aedes*, dengue, global, KAP, morbidity, physicians, treatment

## Abstract

Dengue fever is one of the most important viral infections transmitted by *Aedes* mosquitoes and a major cause of morbidity and mortality globally. Accurate identification of cases and treatment of dengue patients at the early stages can reduce medical complications and dengue mortality rate. This survey aims to determine the knowledge, attitude, and practices (KAP) among physicians in dengue diagnosis and treatment. This study was conducted among physicians in Turkey as one nonendemic country and Bangladesh, India, and Malaysia as three dengue-endemic countries. The dosing frequencies, maximum doses, and contraindications in dengue fever were examined. The results found that physicians from Bangladesh, India, and Malaysia have higher KAP scores in dengue diagnosis and treatment compared to physicians in Turkey. This may be due to a lack of physician’s exposure to a dengue patient as Turkey is considered a nonendemic country. This assessment may help establish a guideline for intervention strategies among physicians to have successful treatment outcomes and reduce dengue mortality.

## 1. Introduction

Dengue fever is a mosquito-borne viral disease that is spreading at a rapid rate across the globe, with an estimated 390 million dengue infections per year [1,2]. Dengue virus is transmitted via the bite of an infected female *Aedes* mosquito [3,4]. The four main dengue virus (DENV) serotypes responsible for causing the infection are DENV-1, DENV-2, DENV-3, and DENV-4 [5]. The infection is primarily characterized by fever, nausea, rash, vomiting, and aches or pains typically behind the eyes, muscles, joints, or bones [6,7]. These symptoms usually last for 2–7 days [1,3,7]. Sometimes, the infection becomes severe and causes internal bleeding, resulting in dengue hemorrhagic fever (dengue with a warning sign). In more serious conditions, it may lead to the development of dengue shock syndrome or even death [8].

Dengue fever is predominantly seen in tropical and subtropical regions. Asian countries represent 70 percent of the global disease burden of dengue fever [1,2]. In India, dengue fever is endemic in almost all the states and is the leading cause of hospitalization [9]. Many decades earlier, it had a predominant urban distribution but is now reported in rural areas as well [9]. Dengue fever has been a major public health concern in Bangladesh since 2000 [10]. During the 2000 outbreak, DENV-3 was the predominant serotype [11]. After 2000, there has been an outbreak every year with different severity and amplitude [10]. The first major dengue fever outbreak in Malaysia was recorded in 1973 [12]. An increase in dengue cases was seen between 1980–2000 due to rapid population and infrastructure growth [13]. In the 21st century, a seven-fold increase was seen in cases from 2000–2010 [13]. Turkey is not considered a dengue-endemic country anymore because the last severe dengue fever outbreak in Turkey was reported in the late 1920s [14]. This outbreak has infected more than 1 million and caused the death of nearly 1500 people [14]. However, the existence of *Ae. albopictus* was first recorded in Turkey (Edirne) in 2011 [15]. Recently in 2015, the reoccurrence of *Ae. aegypti* and wide distribution of *Ae. albopictus* populations were reported in the Black Sea region [16]. Reasons for the rapid transmission of dengue fever in these countries include (1) temperature and climatic conditions that play a key role in viral replication, infective periods, and viral-vector survival [17,18,19]; (2) globalization, international travel, and trade that increases the risk of epidemic transmission [20]; and (3) increasing population, socioeconomic and behavioral factors, such as not using insect repellents, water storage, and disposal techniques, poor sanitation, and low income, in the countries located in the tropical and subtropical regions [19,21].

After the onset of dengue fever, the virus can be isolated and detected in the circulating blood cells, serum, and plasma within 4–5 days [22]. Antibodies against the dengue virus are detectable in 99 percent of the patients within 10 days of disease onset, with serologic tests [22]. Nonetheless, an array of laboratory diagnostic methods are available for detecting dengue fever. The choice of the diagnostic method depends on the patient’s signs and symptoms, time of sample collection, costs, and available laboratory facilities [23]. There is no effective vaccine for dengue fever. Instead, it is usually managed based on the clinical signs and symptoms of the infected patients [24]. Basic steps of dengue management include (1) treating the early febrile phase of dengue fever with the WHO recommended nonsteroidal anti-inflammatory agents (NSAIDs); (2) initiating oral or intravenous fluid therapy to stabilize hemodynamic status, transfusing blood in the case of severe bleeding; and (3) closely monitoring other clinical parameters, such as platelet count, hematocrit, and urine output [22].

The key to favorable clinical outcomes in dengue fever is early detection and effective management of symptoms by health care professionals (HCPs) [6,25]. Thus, the knowledge of HCPs plays a vital role in the diagnosis and management of dengue fever, thereby reducing its complications, morbidity, and mortality rates. A few studies conducted to assess the knowledge, attitudes, and practices (KAP) of HCPs in the management of dengue fever showed good knowledge among physicians regarding dengue fever [26,27,28]. Another KAP study showed that only a moderate level of knowledge was observed in 49 percent of the HCPs, with room for improvement in their level of attitude and practice concerning dengue fever (dengue without warning sign) prevention [29]. One study reported that almost every physician in the sample knew that aspirin and non-steroidal anti-inflammatory drugs must be avoided during dengue fever or shock syndrome (severe dengue) [30]. However, other studies reported that physicians lacked knowledge about clinical signs, symptoms, and treatment for dengue patients [31]. One published study anticipated there were significant variations in clinical practice by physician’s age group and practice setting [27]. At the same time, other published studies reported that the KAP of healthcare professionals for dengue fever was not well established [29,30,32]. To our best knowledge, there were no KAP studies conducted among physicians in India, Bangladesh, Malaysia, and Turkey.

Due to the rapid geographic expansion of *Aedes* mosquitoes [33,34,35,36], it is important to have proper knowledge and practices among physicians for dengue fever diagnosis and treatment. Several studies conducted in developing countries revealed that the diagnostic facilities for dengue fever were poor, especially in the rural regions [7,37,38]. Because of this, physicians need to have better knowledge and understanding of dengue fever for early detection and management. The assessment of physicians’ KAP level plays a significant role in dengue diagnosis and treatment. Previous studies that assessed the KAP levels of physicians were based on a small geographic area [28,32].

There were no studies on KAP on physicians (to our best knowledge) about dengue diagnosis and treatment in these four countries. Malaysia is a dengue-holoendemic country, India and Bangladesh are hyperendemic countries, and Turkey is a nonendemic country. The current study aims to understand the challenges for treatment, management, and diagnosis of dengue fever among physicians by evaluating their KAP. In the current study, a survey was conducted among physicians from one nonendemic and three dengue-endemic countries to have an in-depth investigation on the challenges for physicians to diagnose dengue fever accurately. The questionnaire for this study was adapted from previous studies [26,27,31,32]. However, this study also emphasized the dosing frequencies, maximum doses, and contraindications in dengue fever.

## 2. Methods

***Participants*:** We obtained e-mail addresses of 583 physicians across all four countries, and the survey link was e-mailed to physicians from December 2020 to January 2021. Physicians who hold a degree for Bachelor of Medicine, Bachelor of Surgery (MBBS), or Doctor of Medicine (MD) were included in this study. A total of 227 (38.94%) physicians completed the survey. Among the participants, 67 are currently practicing in Bangladesh, 61 physicians are located in India, and 41 and 54 are practicing in Malaysia and Turkey, respectively.

***Procedure*:** The current study was designed as a web-based cross-sectional survey. We targeted the primary care physicians and general practitioners that deal with dengue fever treatment and management in hospital setting. A total of three gentle reminders were sent to the physicians biweekly, reminding them to complete the survey. The survey was e-mailed to physicians in Bangladesh, India, Malaysia, and Turkey. The e-mail addresses of the physicians were obtained by contacting the respective medical associations of these countries. Four of the 227 respondents were excluded from the final analysis as they indicated that they were no longer in clinical practice.

***Questionnaire preparation*:** For this study, a questionnaire was adapted from previous studies, which was developed based on the 2009 WHO guidelines on dengue fever [39]. The questionnaire was pilot tested in Bangladesh, India, Malaysia, and Turkey to assess the level of difficulty and lucidity. The questionnaire was shared with a convenience sample of 10 physicians, and their inputs were considered. Finally, a 31 item questionnaire was developed. The approximate time to fill out this questionnaire was 15 min.

### Measures

***Knowledge*:** The knowledge section of the questionnaire includes questions on clinical signs and symptoms, diagnostic tests, admission and discharge criteria for patients, treatment options and plans, and complications of dengue fever. This domain had a total of 19 questions. All the questions in this section were asked in the form of multiple choices. Correct responses were given a score of 1, and incorrect responses were given a score of 0. The overall knowledge score ranged from 0 to 19.

***Attitude*:** A total of four questions were asked to assess the attitude of physicians toward dengue fever. For questions in this section, a score of 0 was given if the respondent chose options of 1 or 2 on the scale, and a score of 1 was assigned if the respondent chose one among the options 3–5. The overall attitude score ranged from 0 to 4. The attitude of the physicians was considered positive if they scored greater than or equal to 2 (≥50%), and it was considered negative if they scored less than 2 (<50%).

***Practice*:** The practice section includes eight questions in total. The physicians were asked questions related to their practice in both clinical and community setups. Similar to the scoring in the knowledge section, a score of 0 was given to incorrect responses, and a score of 1 for correct responses. In this section, physicians can score a maximum of 8 and a minimum of 0. Physicians with a score of 4 or above (≥50%) were considered to have good practices, and scores less than 4 were categorized to have poor practices.

***Statistical analysis*:** A combined score of 16 or greater (>50%) across all the three domains were considered as a good KAP level, and any scores less than 50% were considered as poor KAP levels [40]. The reason for summing the scores in three domains is that practices and attitude of physicians depend on their knowledge on dengue fever. Few questions in the questionnaire had “Don’t know” as one of the options. All the “Don’t know” responses were pooled with wrong answers and were scored 0. This is because all the respondents who chose “Don’t know” were uncertain about the answer, or they do not know the answer.

Logistic regression analysis was used to identify the predictors of the KAP level of dengue fever among the physicians. Independent variables were the country of residence, clinical experience (in years) in treating dengue fever, age, and sex of the physician. The experience here is the clinical experience of physicians in diagnosing and treating dengue fever. Based on the KAP score, a dependent variable called KAP level (good vs. poor) was introduced. We have used a binary logistic regression with sex, country, clinical experience, and age as predictors. The responses of the physicians were classified based on the place of residence (country) [41]. The analysis was done using SAS (v. 9.4; SAS Institute Inc., Cary, NC, USA), and the graphs were produced using Microsoft Excel (Microsoft Corp., Redmond, WA, USA).

**Ethical approval:** This study was reviewed by the North Texas regional IRB and approved. No incentives were offered to physicians for completing the survey. A consent script was included before beginning the survey, and the participants were given an option to “disagree” for not participating in the survey.

## 3. Results

In this study, more than half of the physicians included in this study were female (53.36%), and most of them belonged to the age group <30 years (58.30%). Nearly 31.84% of the physicians reported their clinical experience as 2–5 years, followed by <2 years (30.49%) (Table 1).

***Knowledge domain*:** High-scoring questions included identification of the correct treatment options for the prevention of dengue fever (90.13%) and recognizing that nonsteroidal anti-inflammatory drugs (NSAIDs) and corticosteroids are contraindicated in dengue fever (83.86%) (Table 2). The lowest scoring questions in this section included determining the criteria for initiating intravenous (IV) fluid therapy (5.83% correct), followed by identification of reasons for NSAID and corticosteroid contraindication (11.21% correct).

The overall knowledge about the signs and symptoms of dengue fever was found to be moderate (43.5%) among the physicians, and it varied based on their location (country) of practice (Table 2). Less than a quarter (21.52%) of the respondents identified all the possible diagnostic tests of dengue fever accurately, although most of the respondents (67.26%) reported that diagnostic facilities were available in their clinical settings for the detection of dengue fever. Moreover, only 23.32% of the physicians could identify the parameters (white blood cells, platelet cells, and hematocrit) in a complete blood count test. About one third (28.25%) of the respondents consider a platelet count of <50,000 cells as a criterion for hospital admission for dengue fever. Only 12.11% of the physicians correctly identified that parameters such as platelet cells, hematocrit, and white blood cells determine possible plasma leakage. The majority of the physicians (90.13%) had excellent knowledge regarding the treatment of dengue fever, which includes management with paracetamol, IV fluid therapy, and blood transfusion. When asked about the maximum daily adult dose of paracetamol for dengue fever, nearly 44% of the physicians responded correctly as 4 g per day. However, only 12.11% of the physicians were aware of the possible adverse effects associated with paracetamol such as rashes, nausea, vomiting, constipation, and increased liver enzyme counts (Table 2).

Most respondents (83.86%) were aware that NSAIDs and corticosteroids are contraindicated in patients with dengue fever. However, only 11.21% of the physicians responded accurately when asked about the reason for the contraindication of NSAIDs and steroids. Only 5.83% of the physicians were able to identify the criteria for initiating IV fluid therapy. However, most (73.09%) identified that IV fluids act by replacing the lost plasma that occurs due to increased vascular permeability. Nearly half (47.98%) of the physicians appropriately recommended that administration of IV crystalloid solutions would restore the blood pressure in case of dengue shock syndrome. Patients with dengue fever seek hospital care at any point of illness; therefore, it is important to understand the admission and discharge criteria for these patients. More than half (51.57%) of the physicians accurately identified the criteria for discharge for dengue patients (Table 2).

Three quarters (74.89%) of the total respondents correctly identified that dengue hemorrhagic fever and dengue shock syndrome are the possible complications associated with dengue fever. Nearly 36% of the overall respondents appropriately identified that hematemesis, dark-colored stools, and bleeding gums are the early signs of dengue hemorrhagic fever. At the same time, only 27.80% of the physicians accurately determined that hypotension and tachycardia are the early indicators of dengue shock syndrome (Table 2).

***Attitude domain*:** Almost half (48.88%) of the total respondents were “confident” in diagnosing a suspected case of dengue fever on a scale of 1–5 (Figure 1a). However, 8.97% of the respondents were “not confident at all” in diagnosing dengue fever. Nearly 87 (39%) respondents were confident in treating a patient with dengue fever, and physicians from India were more confident compared to physicians in Bangladesh, Malaysia, and Turkey (Figure 1b). When asked about the efficiency of their current clinical practice setting in handling dengue fever cases (Figure 1c), 21 (38.89%) physicians from Turkey responded that they were not confident in handling dengue cases, with their currently available clinical practice settings, whereas physicians from other countries were fairly confident about their clinical practice setting. Figure 1d explains the attitudes of physicians on how effective their respective government is handling dengue fever cases. In general, most (34.98%) of the respondents were not sure, whereas 29.63% of physicians from Turkey responded that their government is not that efficient in handling dengue cases.

***Practice domain*:** Table 3 shows that 163 physicians (73.09%) always advise their patients to avoid NSAIDs such as aspirin and ibuprofen in suspected cases of dengue fever. When asked about the recommendation of prophylactic IV fluid therapy in patients with suspected dengue fever, we received a mixed response from the physicians. Nearly 40% of the physicians do not recommend prophylactic IV fluid therapy; however, 37.67% recommend this therapy for the management of suspected dengue fever. Almost 80% of the respondents revealed that they do not prefer prophylactic platelet transfusion in suspected patients with dengue fever, which is considered good practice. Most of the physicians (87.89%) often look for skin rashes if they suspected dengue fever, while 4.48% of them do not consider this as a clinical manifestation. Similarly, most physicians (90.58%) are inclined to monitor blood pressure and complete blood counts daily in suspected patients with dengue fever. Nearly 94% of the respondents answered that they avoid administering intramuscular (IM) injections to patients with dengue fever. This is considered a fair practice because IM injections may cause hematomas. Almost 65% of the physicians practice handing out homecare cards, which assists them by explaining the practices and cautions that should be adopted for the dengue patients.

***Determinants of KAP score among physicians*:** Regarding KAP scores, 69% of the participants achieved at least 50% on the overall score of good KAP. The study found increased log odds of having a good KAP score if the respondent is from Bangladesh (β: 2.29; 95% CI: 1.46–3.13), India (β: 2.56, 95% CI: 1.67–3.46), and Malaysia (β: 3.59; 95% CI: 2.27–4.91) compared to respondents in Turkey. The study also found increased log odds of having a good KAP score if the respondent has 5–10 years of clinical experience in treating dengue fever (β: 2.32, 95% CI: 0.71–3.92) and greater than 10 years (β: 2.38, 95% CI: 0.63–4.14) compared with clinical experience of fewer than 2 years (Table 4). The country of the physicians and clinical experience of >5 years in treating dengue fever was found to be significant with the KAP scores of the physicians.

## 4. Discussion

This survey conducted among the physicians who were involved with dengue treatments from Bangladesh, India, Malaysia, and Turkey demonstrated the need for minimizing the knowledge gaps and also improving the clinician’s ability to diagnose and management of dengue fever. Irrespective of their country of practice, many physicians failed to identify the scenario for initiation of IV fluid therapy and the reasons for NSAIDs and corticosteroid contraindication in dengue fever. Fair knowledge on these topics is vital for effectively managing dengue fever and preventing complications associated with it. In addition, the majority of physicians failed to identify the parameters of incomplete blood count that helps to determine plasma leakage.

The mean KAP score of the physicians in Turkey was found to be 11.7, whereas, for those practicing in Bangladesh, India, and Malaysia, it was found to be 16.9, 18.03, and 18.46, respectively. The relatively low score which was observed among the physicians from Turkey could be because there are no active dengue cases in Turkey [14]. In addition, *Aedes aegypti* was reported in the Turkish Mosquito Fauna Checklist in 1959, but it was confirmed that it was not found any more in 2001 [39]. However, recent arboviral screening in northeastern Turkey revealed a territorial expansion of *Aedes* species, which is alarming [40]. Therefore, continuous medical training on dengue fever management and sound clinical knowledge is still essential among the physicians practicing in Turkey.

Despite the lack of evidence, the practice of prophylactic platelet transfusion is reported among 20% of the respondents, which is lower when compared to the results obtained in a previous study conducted among the physicians in Puerto Rico [31]. Nevertheless, the practice of prophylactic platelet transfusion is not recommended by the WHO guidelines [22]. In addition, recent studies revealed that prophylactic platelet transfusion does not improve platelet count. It might also lead to several adverse reactions [42] and an increased duration of hospital stay, which further leads to an increase in the total cost of treatment [43]. In the current study, most physicians failed to identify all the possible adverse effects of paracetamol. This could be because paracetamol is relatively safe when taken within the therapeutic dosage limits (<4 g per day in adults) and also because physicians tend to perceive paracetamol as less risky when compared to other analgesics or antipyretics [44].

In addition, it was observed that physician’s knowledge on laboratory diagnosis of dengue fever was low. Only 21.52% of the respondents were able to identify all the diagnostic tests available for dengue fever. However, a study conducted among the physicians practicing in Puerto Rico [31] revealed that only 6% of them knew about the laboratory tests used to diagnose dengue fever. It is important to have a strong knowledge of the diagnostic parameters because early diagnosis of dengue fever decreases the morbidity and mortality rates associated with it [26,42,45]. Even though the WHO guidelines [22] do not recommend the use of corticosteroids, 13% of the respondents do not have the practice of advising patients to avoid steroids when diagnosed with dengue. It is important to instruct patients to avoid steroids because they are likely to increase the risk of upper gastrointestinal bleeding and stress ulceration [46]. In addition, a randomized controlled trial revealed that there is no sufficient evidence to justify the use of corticosteroids in the management of dengue shock syndrome [47].

With the increasing global burden of dengue infection and rapid geographic expansion of *Aedes* mosquitoes due to climate change and habitat suitability, half of the world population is now living at risk. Our study also confirmed that physicians in Turkey need more training on dengue diagnosis and treatment, which is still known as a non-dengue-endemic country but with reported dengue infections. This finding can be implied to other nonendemic countries where the dengue burden is still very low but with reported existence of *Aedes* mosquitoes.

Although the current study provides data from four countries regarding the KAP of physicians in the management of dengue fever, it does have a few limitations. Firstly, our survey relied on self-reported data from clinical practitioners. As a result, the accuracy of the information is not known. Past studies revealed that clinicians tend to over-report their compliance with the standard therapeutic guidelines [31,48]. Our survey had slightly more female participants when compared to males, and the possible reason could be that females are more willing to participate in online studies than males [49]. Most of the respondents in our study are from the age group <30 years, which reveals that the response rate from the senior physicians is low. This might be because the data collection was done in the middle of the COVID-19 pandemic when the physicians were involved with COVID-19 managements, and the responses were mainly given by younger practitioners who had a willingness to participate in this survey. Alternately, young doctors are more internet savvy and familiar with online survey. Finally, nonresponse bias might affect the results. However, our study reported a response rate of 38%, which is higher compared to the previous studies [27,31,50], and revealed new knowledge about dosing frequencies, maximum doses, and contraindications in dengue fever among physicians.

## 5. Conclusions

Despite dengue fever being endemic in Bangladesh, India, and Malaysia, the clinical knowledge of physicians practicing in these countries is not optimal. This calls for a continued and sustained clinical training on dengue fever management. This study also confirms lower KAP scores among physicians in Turkey where the cases are limited. Findings from this survey can be used to design educational interventions or a training course on dengue fever management, which can improve physician’s knowledge and practice in all countries that have reported dengue infections and the existence of *Aedes* mosquitoes. Future studies need to assess the impact of educational interventions on physician’s knowledge and clinical management of dengue fever.

## Figures and Tables

**Figure 1 behavsci-11-00105-f001:**
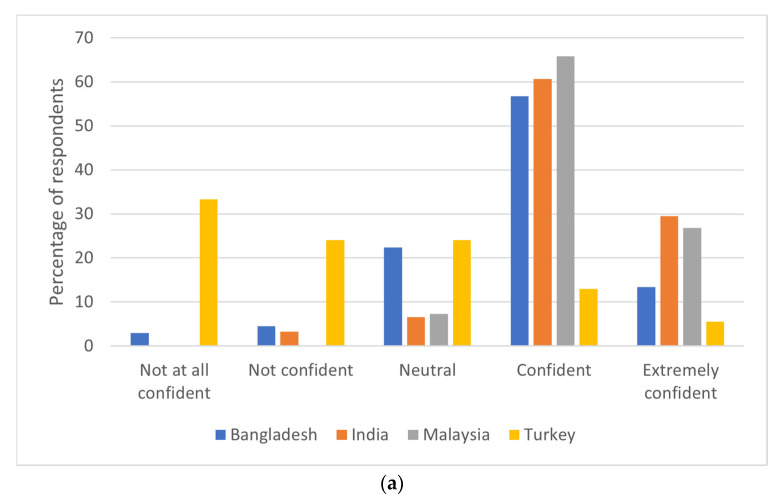
(**a**) Respondents’ confidence in diagnosing dengue by country of practice. (**b**) Respondents’ confidence in treating dengue by country of practice. (**c**) Respondents’ clinical practice setting efficiency by country of practice. (**d**) Government efficiency in handling dengue cases by country of practice.

**Table 1 behavsci-11-00105-t001:** Socio-demographic characteristics of the study population.

Demographics	n (n = 223)	Percentage (%)
**Gender**		
Male	104	46.64
Female	119	53.36
**Place of current residence (Country)**		
Bangladesh	67	30.04
India	61	27.35
Malaysia	41	18.39
Turkey	54	24.22
**Age group**		
<30	130	58.30
31–40	61	27.35
41–50	21	9.42
>50	11	4.93
**Years of clinical experience**		
<2 years	68	30.49
2–5 years	71	31.84
5–10 years	35	15.70
>10 years	49	21.97

**Table 2 behavsci-11-00105-t002:** Physician’s percentage of correct responses by country of practice.

Variable	Bangladesh n = 67 (%)	India n = 61 (%)	Malaysia n = 41 (%)	Turkey n = 54 (%)	Overall n = 223 (%)
Signs and symptoms	27 (40.30)	25 (40.98)	18 (43.9)	27 (50)	97 (43.50)
Diagnostic tests	15 (22.39)	20 (32.79)	6 (14.63)	7 (12.96)	48 (21.52)
Parameters in complete blood count	6 (8.96)	14 (22.95)	26 (63.41)	6 (11.11)	52 (23.32)
Platelet count for hospital admission	19 (28.36)	20 (32.79)	11 (26.83)	13 (24.07)	63 (28.25)
Plasma leakage	10 (14.93)	11 (18.03)	3 (7.32)	3 (5.56)	27 (12.11)
Treatment option	67 (100)	60 (98.36)	40 (97.56)	34 (62.96)	201 (90.13)
Paracetamol dosage frequency	16 (23.88)	29 (47.54)	21 (51.22)	9 (16.67)	75 (33.63)
Paracetamol maximum dosage	22 (32.84)	28 (45.90)	24 (58.54)	25 (46.30)	99 (44.39)
Paracetamol adverse effects	6 (8.96)	11 (18.03)	1 (2.44)	9 (16.67)	27 (12.11)
NSAIDs, corticosteroids contraindication	66 (98.51)	55 (90.16)	40 (97.56)	26 (48.15)	187 (83.86)
Reason for NSAIDs/steroids contraindication	8 (12.12)	12 (20.34)	3 (7.32)	2 (4.44)	25 (11.21)
Initiation of IV fluid therapy	2 (2.99)	2 (3.28)	8 (19.51)	1 (1.85)	13 (5.83)
Role of IV fluid therapy	59 (88.06)	50 (81.97)	31 (75.61)	23 (42.59)	163 (73.09)
IV fluid in restoring blood pressure	33 (49.25)	33 (54.10)	21 (51.22)	20 (37.04)	107 (47.98)
Initiation of blood transfusion	10 (14.93)	9 (14.75)	4 (9.76)	8 (14.81)	31 (13.90)
Discharge criteria	34 (50.75)	34 (55.74)	10 (24.39)	37 (68.52)	115 (51.57)
Complications of dengue fever	54 (80.6)	53 (86.89)	34 (82.93)	26 (48.15)	167 (74.89)
Early signs of dengue hemorrhagic fever	21 (32.31)	27 (44.26)	17 (42.50)	15 (28.30)	80 (35.87)
Early signs of dengue shock syndrome	16 (25.00)	19 (31.15)	19 (46.34)	8 (14.81)	62 (27.80)

**Table 3 behavsci-11-00105-t003:** Dengue practices of physicians by country of practice.

Variable	Bangladesh n = 67 (%)	India n = 61 (%)	Malaysia n = 41 (%)	Turkey n = 54 (%)	Overall n = 223 (%)
**Avoid NSAIDs**					
Yes	56 (83.58)	44 (72.13)	32 (78.05)	31 (57.41)	163 (73.09)
No	6 (8.96)	5 (8.20)	7 (17.07)	11 (20.37)	29 (13.00)
Sometimes	5 (7.46)	12 (19.67)	2 (4.88)	12 (22.22)	31 (13.90)
**Prophylactic IV Fluids**					
Yes	21 (31.34)	28 (45.90)	8 (19.51)	27 (50.00)	84 (37.67)
No	34 (50.75)	21 (34.43)	17 (41.46)	17 (41.46)	89 (39.91)
Sometimes	12 (17.91)	12 (19.67)	16 (39.02)	10 (39.02)	50 (22.42)
**Prophylactic Platelet Transfusion**					
Yes	6 (8.96)	3 (4.92)	0 (0.00)	3 (5.56)	12 (5.38)
No	56 (83.58)	46 (75.41)	36 (87.80)	40 (74.07)	178 (79.82)
Sometimes	5 (7.46)	12 (19.67)	5 (12.20)	11 (20.37)	33 (14.80)
**Skin Rashes**					
Yes	62 (92.54)	60 (98.36)	36 (87.80)	38 (70.37)	196 (87.89)
No	1 (1.49)	0 (0.00)	1 (2.44)	8 (14.81)	10 (4.48)
Sometimes	4 (5.97)	1 (1.64)	4 (9.76)	8 (14.81)	17 (7.62)
**Blood pressure and CBC**					
Yes	62 (92.54)	60 (98.36)	41 (100)	40 (74.07)	202 (90.58)
No	1 (1.49)	0 (0.00)	0 (0.00)	9 (16.67)	11 (4.93)
Sometimes	4 (5.97)	1 (1.64)	0 (0.00)	5 (9.26)	10 (4.48)
**IM Injections**					
Yes	1 (1.49)	0 (0.00)	0 (0.00)	8 (14.81)	9 (4.04)
No	66 (98.51)	58 (95.08)	41 (100)	44 (81.48)	209 (93.72)
Sometimes	0 (0.00)	3 (4.92)	0 (0.00)	2 (3.70)	5 (2.24)
**Health promotion programs**					
Yes	28 (41.79)	21 (34.43)	22 (53.66)	7 (12.96)	78 (34.98)
No	15 (22.39)	14 (22.95)	4 (9.76)	34 (62.96)	67 (30.04)
Sometimes	24 (35.82)	26 (42.62)	15 (36.59)	13 (24.07)	68 (30.49)
**Home care cards**					
Yes	44 (65.67)	49 (80.33)	37 (90.24)	15 (27.78)	145 (65.02)
No	13 (19.40)	3 (4.92)	2 (4.88)	28 (51.85)	46 (20.63)
Sometimes	10 (14.93)	9 (14.75)	2 (4.88)	11 (20.37)	32 (14.35)

**Table 4 behavsci-11-00105-t004:** Predictors of KAP (1 = adequate vs. 0 = inadequate) of dengue diagnosis and treatment among physicians.

Variable	β Estimate	95% CI	OR	*p*-Value
**Sex**				
Female	−0.14	(−0.89, 0.61)	0.87	0.7209
(reference = Male)				
**Country**				
Bangladesh	2.29	(1.46, 3.13)	9.91	<0.0001 *
India	2.56	(1.67, 3.46)	12.99	<0.0001 *
Malaysia	3.59	(2.27, 4.91)	36.19	<0.0001 *
(reference = Turkey)				
**Experience**				
2–5 years	0.93	(−0.12, 1.97)	2.53	0.0818
5–10 years	2.32	(0.71, 3.92)	10.14	0.0048 *
>10 years	2.38	(0.63, 4.14)	10.83	0.0078 *
(reference = <2 years)				
**Age**				
31–40 years	−0.50	(−1.57, 0.57)	0.61	0.3628
41–50 years	−1.73	(−3.60, 0.13)	0.18	0.0690
>50 years	0.45	(−1.90, 2.80)	1.57	0.7060
(reference = <30 years)				

OR: odds ratio; CI: confidence intervals; *: significant at *p* < 0.05.

## Data Availability

The data underlying this article will be shared at reasonable request to the corresponding author due to privacy concerns of physicians who responded to the survey.

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
