# Peer review of "Assessment of Knowledge, Attitudes, and Practices Regarding Dengue among Physicians: A Web-Based Cross-Sectional Survey"

_behavsci, 2021, doi:10.3390/bs11080105_

Round 1

Reviewer 1 Report

Comments on Behavioral Sciences manuscript “Assessment of knowledge, attitudes, and practices regarding dengue among physicians: a Web-based cross-sectional survey.”

General comment: The reported study was aimed at determining the knowledge, attitude, and practices (KAP) among 20 physicians in dengue fever diagnosis and treatment. This study was conducted among physicians in Turkey as one non-endemic country and Bangladesh, India, and Malaysia as three endemic countries.  The study appears to provide new and unique information that could be helpful to promote better healthcare services in the said countries.  Yet, deficiencies in the study and the manuscript severely limit the study’s potential contribution.  Suggestions below might help to address the deficiencies.

Major issue:

  1. The dengue fever situations in Bangladesh, India, and Malaysia should be discussed in the Introduction section, together with situations in Turkey, to provide a context and rationale for including and contrasting these countries.  Also, it may be useful to move Lines 269-276 to the Introduction section to provide background information on Turkey.  This may be particularly useful because the Introduction section currently only mentions that Turkey has not seen a dengue fever outbreak for almost 100 years (Line 45).  If Turkey had not seen any more recent regional problems, then it would have made less sense to contrast it with the other three countries.
  2. It is not clear whether the logistic regression (Table 3) included country of residence as a predictor even though this predictor is mentioned in the methods section (Line 148).  (1) Given the aim of the study, this predictor could very well be the most important insight of the analysis – whether and how prevalence/experience of dengue fever in a country predicts its physicians’ KAP as well as how differences, if any, may be addressed to promote public health.  (2) Though country differences are discussed extensively in the results section and mentioned in the Abstract (Lines 24-25), any conclusions on country differences should be based on the logistic regression, which controls for non-country factors, rather than contingency tables (Tables 1 and 2).

Other issues:

  1. Literature review (Lines 70-85): it would be useful to identify the studies, if any, that have investigated KAP in each of the four countries to clarify what this study can contribute to the literature.
  2. Lines 88-90: it would be useful to clarify why better physician KAP is useful when limitations in diagnostic facility is another factor that may have contributed to the dengue problem in these countries.
  3. Lines 94-99: it would be useful to expand on the rationale behind selecting and contrasting the four countries.
  4. Lines 102-107: were the physicians of the same specialty or of different specialties (what are they)>
  5. Lines 141-143: please explain the rationale behind summing scores in the three domains.  What are the assumptions behind summing?  For example, is a physician who scored 16 or higher in the knowledge domain but 0 in the other two domains “better” or “more adequate” than another physician who scored 10 in knowledge, 2 in attitude, and 3 in practice? 
  6. Lines 315-317: how is the covid-19 pandemic related to age differences in survey response?
  7. Line 318: the low KAP among Turkey physicians is a finding, but not a limitation, of the study.

Author Response

Respected reviewer,

We thank you for your valuable comments. The manuscript was updated and highlighted in Turquoise with corrections. Please find our responses below:

The dengue fever situations in Bangladesh, India, and Malaysia should be discussed in the Introduction section, together with situations in Turkey, to provide a context and rationale for including and contrasting these countries.  Also, it may be useful to move Lines 269-276 to the Introduction section to provide background information on Turkey.  This may be particularly useful because the Introduction section currently only mentions that Turkey has not seen a dengue fever outbreak for almost 100 years (Line 45).  If Turkey had not seen any more recent regional problems, then it would have made less sense to contrast it with the other three countries.

Response: We thank the reviewer for their valuable suggestion. As suggested, the scenario of dengue fever in Bangladesh, India and Malaysia was discussed in the Introduction section (lines 43-56). Also, as suggested, the lines explaining the background information on Turkey were moved to Introduction section. The reason for selecting Turkey as one of the countries is to find the difference in KAP levels of physician from a non-endemic country with that of endemic countries.

It is not clear whether the logistic regression (Table 3) included country of residence as a predictor even though this predictor is mentioned in the methods section (Line 148).  (1) Given the aim of the study, this predictor could very well be the most important insight of the analysis – whether and how prevalence/experience of dengue fever in a country predicts its physicians’ KAP as well as how differences, if any, may be addressed to promote public health.  (2) Though country differences are discussed extensively in the results section and mentioned in the Abstract (Lines 24-25), any conclusions on country differences should be based on the logistic regression, which controls for non-country factors, rather than contingency tables (Tables 1 and 2).

Response: Table on the logistic regression was corrected and updated with country of residence being a predictor. Table 4 is the updated version on logistic regression and the conclusion has accommodated the findings.

Literature review (Lines 70-85): it would be useful to identify the studies, if any, that have investigated KAP in each of the four countries to clarify what this study can contribute to the literature.

Response: We thank the reviewer for their comment. However, there were no KAP studies on physicians (to our best knowledge) that were conducted in these 4 countries. (91-94).

Lines 88-90: it would be useful to clarify why better physician KAP is useful when limitations in diagnostic facility is another factor that may have contributed to the dengue problem in these countries.

Response: Thank you for the comment. Since diagnostic facilities were limited in rural parts of countries like India and Bangladesh, better physician KAP would help in early detection and management of dengue fever based on the symptoms which ultimately improves the patient quality of life. We explained it in lines 93-100.

Lines 94-99: it would be useful to expand on the rationale behind selecting and contrasting the four countries.

Response: We thank the reviewer for their comment. The reason behind selecting these countries, is that we have chosen India and Bangladesh as holoendemic countries, Malaysia as hyperendemic country and Turkey as a non-endemic country. Our aim was to compare the KAP levels of physicians among these countries (lines 102 – 104).

Lines 102-107: were the physicians of the same specialty or of different specialties (what are they)?

Response: We targeted all the primary care physicians and general practitioners whoever are dealing with Dengue fever treatment in hospitals and clinical settings. (line 120).

Lines 141-143: please explain the rationale behind summing scores in the three domains.  What are the assumptions behind summing?  For example, is a physician who scored 16 or higher in the knowledge domain but 0 in the other two domains “better” or “more adequate” than another physician who scored 10 in knowledge, 2 in attitude, and 3 in practice? 

Response: That is a very interesting comment. The reason for summing score in the three domains is because, the practice of physicians does depend on their knowledge on dengue fever(153-154). For example, only physicians with knowledge that NSAID’s are contraindicated in dengue fever will be able to advise patients to avoid them if they are diagnosed with dengue fever. Hence the score was summed in three domains.

Lines 315-317: how is the covid-19 pandemic related to age differences in survey response?

Response: Since the data was collected amidst COVID-19 pandemic, we believe most of the senior physicians were busy in COVID specific duties and hence the response rates were low in senior age group of physicians. In line 337-340 we added few lines “ when the physicians were involved with Covid-19 managements and the responses mainly made from younger practitioners who have willingness to participate in this survey. Alternately, the young doctors are more internet savvy and familiar with online survey”. 

Line 318: the low KAP among Turkey physicians is a finding, but not a limitation, of the study.

Response: We thank you for the suggestion. This line was removed from the limitations.

Reviewer 2 Report

I revised a study titled “Assessment of knowledge, attitudes, and practices regarding dengue among physicians: A web-based cross-sectional survey”.

It is an interesting web-based cross-sectional survey with the aim to understand the challenges for treatment, management and diagnosis of dengue fever among physicians about the important health problem of dengue fever diagnosis or treatments.

The manuscript addresses an issue that is very important and relevant in health care and in particular in the infectious disease field. I have some comments to address to the authors.

It is not clear why the survey was on the 4 countries investigated.

The Table A1: Socio-demographic characteristics of the study population should be the first table.

Statistical analysis: It is not clear the statistical model used to investigate the dependent variable KAP (poor/good) reporting  the independent variables and how they was measured (site of practice or experience).

You reported that you used a regression model with countries as independent variables and on the abstract I read that “The results found that physicians from Bangladesh, India, and Malaysia have higher 24 KAP scores in dengue diagnosis and treatment compared to physicians in Turkey”

On the manuscript you reported : “ Since the responses of the physicians are nested within the country of physicians, a random effect model was used” . I think is better clarify on the methods section, the specific model run to study the predictors of KAP and if they used a logistic regression model with the random effect, it should be reported on statistical analysis and in table 3.

I think is important to clarify what is the factor “experience”

Why did you introduce into the model “age” as a factor? How  did you choose the class distribution of age?  Why the age categories start from 21 years?  The Authors reported  that the participants were “Physicians who hold a degree for Bachelor of Medicine, Bachelor of Surgery (MBBS) 104 or Doctor of Medicine (MD)” , could you better explain the inclusion criteria of the study? Could the participants hold a degree for bachelor of medicine/surgery, have clinical experience and to be  21 years old?

Table 3 I think there was a mistake for the age reference (21-25)

By  the age distribution from Table A1, I suggest to reduce the classes , check the percentage the total is not 100%.

Author Response

Respected Reviewer,

We thank you so much for your valuable comments. The manuscript has been updated and highlighted in Turquoise for your kind reference. Please find the responses below:

It is not clear why the survey was on the 4 countries investigated.

Response: We thank the reviewer for their comment. The reason behind selecting these countries, is that we have chosen India and Bangladesh as holoendemic countries, Malaysia as hyperendemic country and Turkey as a non-endemic country. Our aim was to compare the KAP levels of physicians among these countries (lines: 102 – 104).

The Table A1: Socio-demographic characteristics of the study population should be the first table.

Response: Thank you for the valuable suggestion. We have moved the socio-demographic characteristics table. Please find the updated Table-1.

Statistical analysis: It is not clear the statistical model used to investigate the dependent variable KAP (poor/good) reporting the independent variables and how they was measured (site of practice or experience). You reported that you used a regression model with countries as independent variables and on the abstract, I read that “The results found that physicians from Bangladesh, India, and Malaysia have higher KAP scores in dengue diagnosis and treatment compared to physicians in Turkey”

On the manuscript you reported: “ Since the responses of the physicians are nested within the country of physicians, a random effect model was used”. I think is better clarify on the methods section, the specific model run to study the predictors of KAP and if they used a logistic regression model with the random effect, it should be reported on statistical analysis and in table 3.

Response: We are extremely sorry for the confusion. Table on the logistic regression was corrected and updated with country of residence being a predictor. Table 4 is the revised  version on logistic regression. We have used a binary logistic regression with sex, country, clinical experience and age as predictors (Line 161).

I think is important to clarify what is the factor “experience”. Why did you introduce into the model “age” as a factor? How  did you choose the class distribution of age?  Why the age categories start from 21 years?  The Authors reported  that the participants were “Physicians who hold a degree for Bachelor of Medicine, Bachelor of Surgery (MBBS) or Doctor of Medicine (MD)” , could you better explain the inclusion criteria of the study? Could the participants hold a degree for Bachelor of Medicine/surgery, have clinical experience and to be 21 years old?

Response: Experience here is the clinical experience of physicians in diagnosing , treating and management of  dengue fever. The age categories were now corrected to <30, 30-40, 40-50 and >50. Any physician with a valid medical degree was asked to respond the survey (Lines: 159 – 161).

Table 3: I think there was a mistake for the age reference (21-25). By the age distribution from Table A1, I suggest to reduce the classes, check the percentage the total is not 100%.

Response: We thank the reviewer for pointing this out. Classes for age groups was reduced and the total percentage was corrected.

Round 2

Reviewer 1 Report

Thank you for considering and incorporating my comments.  I think the manuscript is in a good shape.  The only thing that I'd suggest for your reconsideration is the explanation about summing the KAP scores - could you propose a stronger or different justification.  The new text (Lines 156-157) made me wonder if you were implying a weighting scheme for the three components - if knowledge is the basis of practices and attitude, then knowledge should be assigned more weight than practices and attitude rather than treated equally.